# Health-Related Quality of Life of Tuberculosis Patients during the COVID-19 Pandemic in Conakry, Guinea: A Mixed Methods Study

**DOI:** 10.3390/tropicalmed7090224

**Published:** 2022-09-02

**Authors:** Almamy Amara Touré, Aboubacar Sidiki Magassouba, Gnoume Camara, Abdoulaye Doumbouya, Diao Cissé, Ibrahima Barry, Lansana Mady Camara, Abdoul Habib Béavogui, Alexandre Delamou, Vanessa Veronese, Corinne Simone Merle, Hugues Asken Traoré, Adama Marie Bangoura

**Affiliations:** 1National Centre of Training and Research in Rural Health of Mafèrinyah, Forécariah BP 2669, Guinea; 2Department of Public Health, Faculty of Health Sciences and Techniques, Gamal Abdel Nasser University, Conakry BP 1147, Guinea; 3Department of Medical Sciences, Université Kofi Annan de Guinea, Conakry BP 1367, Guinea; 4National Tuberculosis Control Program, Conakry BP 1147, Guinea; 5Centre d’Excellence Africain pour la Prévention et le Contrôle des Maladies Transmissibles (CEA-PCMT), Gamal Abdel Nasser University, Conakry BP 1017, Guinea; 6The Special Programme for Research and Training in Tropical Diseases (TDR), World Health Organization, 1211 Geneva, Switzerland

**Keywords:** tuberculosis, health-related quality of life, COVID-19, Guinea

## Abstract

The COVID-19 pandemic has had a significant impact on all facets of life and has exacerbated many challenges faced by people living with tuberculosis (TB). This study aimed to assess the health-related quality of life (HRQoL) of TB patients in Guinea during the first wave of the COVID-19 pandemic. A mixed methods study was conducted using two validated tools to assess HRQoL and qualitative interviews among TB patients enrolled in treatment at 11 health centers in Conakry, Guinea. Logistic regression was used to identify factors associated with the deterioration of HRQoL. We included 439 participants in the study, among whom 44% and 31% experienced pain and anxiety, respectively. We found that an increase in the number of household size and the distance from participants’ residence to the health centers were significantly associated with lower HRQoL. Qualitative interviews highlighted nutritional and financial issues, which were exacerbated during the COVID-19 pandemic and beliefs that the Guinean Government’s assistance plan was insufficient. This study supports the implementation of specific relief plans for TB patients, which includes nutritional and psychological support, especially those whose movements are limited by travel restrictions, preventing access to TB care, reducing work opportunities and exacerbating financial needs and stress.

## 1. Introduction

Tuberculosis (TB) is one of the leading causes of death due to an infectious disease in low and middle-income countries [1]. The World Health Organization (WHO) estimates that 10 million people fell ill with TB in 2019, including 1.4 million deaths [1]. According to the Guinean National Program of Tuberculosis (NTP), the estimated number of notified cases was 122 in 2020 per 100,000 inhabitants, with a treatment success rate of 72% [2]. However, this notification rate is lower than the WHO national estimates of 176 cases per 100,000 inhabitants, suggesting ongoing challenges with TB screening and detection in Guinea [2]. Such challenges are driven by factors including low coverage of TB services, human resources shortages, and TB cases among children and lack of follow-ups among patients [1]. For those who are diagnosed, the successful management of TB patients in Guinea is impeded by difficulties associated with implementing directly observed treatment (DOT), which requires patients to attend TB clinics daily during throughout their TB treatment [3,4]. Other factors, such as patient characteristics, experiences of stigma and psychological and emotional dysfunction, financial problems, side effects of anti-TB drugs, HIV co-infection and the overall quality of care available to TB patients, also impact treatment adherence and success [5,6,7]. The emergence of COVID-19 has exacerbated many of these challenges, further complicating the management and prevention of TB. According to WHO, a 25–30% global reduction in the number of TB cases reported and treated over three months is estimated to result in an additional 0.2 to 0.4 million additional deaths [1].

Since the beginning of the COVID-19 pandemic in Guinea in March 2020, several mitigation actions have been taken by the Guinean authorities to control the pandemic, which, by September 2020, had resulted in more than 10,000 infected people and 66 deaths [8]. The mitigation measures, enacted under a state of a health emergency, included the closure of places of worship, schools and universities, a ban on gatherings of more than ten people, including tournaments and artistic events, a reduction in the number of passengers on public transportation, mandatory wearing of masks, the closure of land, sea and air borders, specific restrictions on the transport of people and goods, and the suspension of international conferences and symposia. Several relief measures were also introduced, including the payment of electricity and water bills for six months from April 2020, the free distribution of gloves and protective gel and the reduction in some food prices. Despite these mitigation measures, they were commonly regarded as insufficient or unpopular; for example, the closure of places of worship was the first experience of the Guinean people and was reluctantly accepted. The increase in the cost of transport following the public transport restrictions acted as a brake on the movement of patients, and closure of borders limited trade with neighboring countries and the availability of goods. It should be noted that no specific actions have been taken that are focused on people at high risk, such as TB patients, who are more likely to experience more severe negative outcomes from COVID-19 compared to non-COVID-19 -TB patients [9]. COVID-19 and TB share similar symptoms and clinical presentation, including cough and fever [10]. However, unlike TB, the global COVID-19 response has received a high-level of attention from decision-makers, leading to the establishment of a rapid system and a response plan [10].

The first pillar of the WHO’s End TB strategy emphasizes patient-centered care and prevention [11]. Health-related quality of life (HRQoL) relates to a perception or response to physical, social, environmental and mental factors that contribute to a normal life, and plays an important role in optimizing treatment success among TB patients [12]. The existing literature suggests that low levels of HRQoL among TB patients are associated with negative health outcomes, such as physical pain, anxiety, and negative therapy-treatment outcome and psycho-social aspects [13,14]. HRQoL may be influenced by the clinical evolution of TB, with an impaired mental well-being being associated with the early months of treatment [15]. Additionally, emerging evidence has documented the negative impacts of the overall HRQoL during the COVID-19 pandemic, particularly among the elderly [16], people who were infected by COVID-19 [17], and those with pre-existing medical conditions [18,19].

Given the impact of both TB and COVID-19 on quality of life, we hypothesize that the risk of deteriorating HRQoL among TB patients may be worse during the COVID-19 era compared to the pre-COVID-19 period, particularly in a context such as Guinea, where there is a marked lack of support for mitigating the impact of COVID-19 among TB patients. However, little is known about the HRQoL among TB patients in Guinea either before, and more importantly during, the COVID-19 era. The goal of this study was to assess the psychosocial and traumatic impacts of COVID-19 on TB patients’ HRQoL in Guinea, in order to inform appropriate, mitigate and protection measures.

## 2. Materials and Methods

### 2.1. Settings

This study was conducted in Conakry, the capital city of Guinea, with 2,039,725 estimated inhabitants [2]. This study took place among TB patients recruited from 11 TB diagnosis and treatment centers (DTCs) in 3 municipalities within the capital city of Conakry (Matoto, Ratoma and Dixinn), the current epicenter of the COVID-19 pandemic in Guinea and where 60% of national TB cases are reported [2]. TB notifications in Conakry ranged from 378 cases per 100,000 inhabitants in 2018 to 409 cases per 100,000 inhabitants in 2019 [2].

### 2.2. Study Framework Analysis

Given the deterioration in the quality of life of TB patients even outside the context of COVID-19 [20], we hypothesize that the COVID-19 pandemic could led to reductions in the reported HRQoL of TB patients in Guinea. Figure 1 shows a hypothesized problem tree. This problem tree was developed through the discussion between the key actors of Guinea’s national TB program. Three key factors that influence HRQoL emerge from this tree and are as follows:(1)The level of information of the population, which was not suitable to explain the risk of infections for vulnerable people.(2)The weak commitment of the Ministry of Health and civil society actors in treating TB, resulting in a lack of funding, which may exacerbate bottlenecks in the treatment of TB patients.(3)The low level of information from community media on the potential risks of COVID-19 and weak community involvement, which may increase the level of fear and stigma.

Collectively, these three interrelated factors ultimately lead to the worsening of TB patients’ quality of life (Figure 1).

### 2.3. Study Design and Population

We carried out a mixed methods study to explore the perception of the impact of COVID-19 and the resulting restrictive measures established by the Guinean authorities on HRQoL among TB patients. The mixed methods study comprised a quantitative, cross-sectional study and qualitative semi-structured interviews. Any patients (aged 10 to 80 years) currently receiving TB treatment at any of the 11 selected DTCs in Conakry during the study period (1 September to 18 September 2020), and who were willing to participate in the study were eligible for inclusion. Those who did not consent to participate in the study were excluded. Conakry DTCs stand for 60% of the Guinea TB patients, but we randomly selected 11 over 24 (45.83%) based on their list. It should be noted that the study period coincided with the end of the first wave of the pandemic.

### 2.4. Study Procedures and Tools

Two validated psychometric tools were used for the quantitative component. First, the Impact of Event-Revised Scale (IES-R) [21], a test of 22 difficulties that people sometimes experience following a traumatic event that is designed to measure the effect of life stress, daily trauma, and acute stress. We used the validated French version of IES-R [22]. Second, the descriptive system for HRQoL states in adults (EQ-5D-3L), designed to provide a generic and straightforward health measurement for clinical and economic evaluation and measures the following five dimensions of health: mobility, self-care, usual activities, pain or discomfort and anxiety or depression [23]. In addition to these psychometric tools, a quantitative questionnaire was developed to collect the following patient information: age; sex (male/female); source of COVID-19 information; marital status (single, married); municipality of residence of the patients; education level (no formal education, primary, secondary, university, high school), occupation of the patient (unemployed, private employee, civil servant, freelance), number of people in the household, and the distance between the patient’s home and the treatment site (in kilometers). The following TB clinical variables were also collected: date of diagnosis, type of TB (susceptible, resistant), localization (pulmonary, extra-pulmonary), treatment history (new case, previously treated), method of diagnosis (bacteriologically confirmed, clinically diagnosed), treatment start date, type of treatment regimen (susceptible, multidrug-resistant), if multidrug-resistant, type of treatment regimen (short course, long course), HIV status (positive, negative), if HIV positive, antiretroviral treatment commenced (yes, no), and the frequency of appointments for drug supply (in days).

For the qualitative component, a semi-structured interview guide was developed to assess patients’ perceptions of COVID-19 and of the restrictive measures implemented. The interview guide explored the following themes: attitudes towards restrictive and measures, proximity to people affected by COVID-19, and the physical, psychological, social, and economic aspects (income, means of transportation). Interviews were conducted in the participant’s preferred language. Investigators were trained in survey methodology, collection tools, and collection technique during a one-day training session, which allowed the interviewers to master the context of the study and its objectives. All research tools were reviewed by the research team and suggestions for improvement were integrated to ensure clarity and improve the overall tool quality and coherency. A pre-test of the tools was organized to assess the feasibility of the field survey.

### 2.5. Sampling and Recruitment

Consecutive sampling was used to recruit eligible patients into the quantitative study from the 11 DTC study sites. Eligible patients were enrolled during their follow-up visits at each of these DTCs, with the number of patients to be enrolled at a DTC being relatively proportional to the number of TB patients on treatment at the centre. The estimated sample size for the quantitative survey was based on a conservative assumption that 50% of TB patients would have a deterioration of HRQOL based on the experiences and professional opinions of the study team. With the desired accuracy of 5%, the minimum expected size was 384; because of non-response rates, this size was increased by 15% or 441 patients.

Participants for the qualitative component were purposely sampled from patients already recruited into the quantitative study. The study team aimed to recruit approximately thirty participants.

### 2.6. Data Collection

For the quantitative component, patient follow-up agents (equivalent to community health workers) and other investigators were equipped with Android cell phones, which were used to administer the questionnaires at the DTCs closest to the patient. Data were recorded through an open data kit (ODK) application (XLSForm with integration into the ONA platform: https://ona.io/home/ accessed 20 August 2020) and connected to a server deployed for this purpose. For the qualitative interviews, interviewers used Android phones to record qualitative interviews accompanied by note-taking.

### 2.7. Data Management and Analysis

Quantitative data were entered directly into the ODK Platform and checked regularly by study team members. The setup of the variables assured good internal quality control and minimized errors and/or duplicate data. Data were carefully cleaned up before analysis. Descriptive statistics were used to summarize patient socio-demographic and clinical data captured by the questionnaires, while the analysis of psychosocial and traumatic events was carried out according to the tool used. For the EQ-5D-3L, traumatic events were calculated according to the three levels for each indicator (no problem, moderate problems, serious problems). Patients with scores corresponding to levels 2 and 3 (moderate and severe problems, respectively) were regarded as having a negative HRQOL. For the revised Event Impact Scale (IES-R), the proportions of patients that reported symptoms of mild post-traumatic stress disorder (PTSD), moderate PTSD, and severe PTSD were calculated based on sociodemographic and clinical features. All TB patients with a score equal to or less than 32 were considered to have low-stress levels, those with a score of 33–38 as moderate-stress levels and those with a score equal to or greater than 39 as severe-stress levels. Confidence intervals were built around these proportions. Chi-square test was used to compare the dependent variables of EQ-5D-3L and IES-R with sociodemographic and clinical variables. Multinomial logistic regression was used to identify significant associations between the ED-5D-3L variables (pain/discomfort, personal care, and anxiety/depression) and the composite variable, resulting from the recoding of the impact scale of the event with sociodemographic and clinical variables. A backward stepwise regression was used to build the model, using R. All the tests were considered significant, with a risk of α = 0.05.

Recordings of the qualitative interviews were transcribed verbatim in French to facilitate data analysis. A thematic approach was used to develop a coding framework, which was then applied to the transcripts. After a preliminary read of the data, the following two major themes emerged: the socioeconomic aspect of TB and the Government’s response to COVID-19. The following sub-themes were applied to the transcripts: public perception/view of TB patients when they cough, assessment of physical and psychological health conditions, the current economic situation as well as restrictive, punitive and stimulative actions taken by the Government regarding TB patients during the COVID-19 pandemic. Coding was conducted using the package RQDA in R software.

## 3. Results

### 3.1. Quantitative Findings

From 29 August to 17 September 2020, a total of 439 TB patients were included in the study, with a mean age of 35 years, primarily male (61%), and the mean distance between the patient’s home and the treatment site was 6.96, with a range of 2–100 (Table 1). The majority of patients had drug sensitive TB (90%), while the 10% with MDR-TB were mostly on a short regimen. One quarter (24.4%) of participants were HIV positive, among whom 90% reported receiving ARV treatment. The majority of the patient’s treatment started within ≤21 days (91%), and less than half had completed more than three months of treatment (47.4%). Finally, the majority of our patients reported more than 90 days with regard to the supply of anti-TB drugs (Table 2).

### 3.2. Traumatic Events Reported by TB Patients

Pain and discomfort, and anxiety and/or depression were the dominant traumatic events reported among the 439 participants surveyed, with 46.5 and 36.9% of participants reporting either a moderate or serious level of impairment, respectively (Table 3).

In the univariate analysis, the following variables of household size, disease localization, TB diagnostic, distance to treatment site and the frequency of appointments for drug supply (in days) were significantly associated with stress; the following were significantly associated with anxiety/depression: level of education, household size, distance to treatment site, and the frequency of appointments for drug supply (in days); age, household number size and whether or not antiretroviral treatments were significantly associated with pain/discomfort; age and distance to treatment site were significantly associated with self care; age, level of education, disease localization, treatment history, the frequency of appointments for drug supply (in days) and duration of treatment were significantly associated with usual activities and age, household size, disease localization and the frequency of appointments for drug supply (in days) were significantly associated with mobility (Appendix A).

The multivariate analysis identified numerous associations between the domains and the following socio-demographic and clinical variables. Difficulties performing usual activities were positively and significantly associated with elderly participants (OR = 5.42; 95% CI: 1.57–22.2; *p* < 0.05) and patients with drug-resistant TB (OR = 27.7; 95% CI: 6.02–132; *p* < 0.005). Those siginficantly less likely to report difficulites with perfoming usual activities included previously-treated patients (OR = 0.02, CI: 0.00–0.011), HIV-negative patients (OR = 0.52; 95% CI: 0.32–0.87; <0.05) and those who had been on treatment for more than 90 days (OR = 0.56; 95% CI: 0.32–0.98; *p* < 0.05) (Table 4).

For a one unit increase in the household size, the odds of reporting pain and discomfort was 1.05 times greater (OR = 1.05; CI: 1.00–1.11, *p* < 0.05), and patients who were clinically diagnosed were 93% more likely to have pain and discomfort than those diagnosed bacteriologically (OR = 1.93,CI: 1.06–3.54, *p* = 0.031), while patients whose frequency of appointments for drug supply was between 15 and 30 days were 66% less likely to have pain/discomfort than those whose frequency of appointments for drug supply was within 14 days (OR = 0.34,CI: 0.16–0.67, *p* = 0.003) (Table 5).

For anxiety, we found that an increase in one unit in the household size was associated with a higher likelihood of reporting anxiety (OR = 1.08, CI: 1.03–1.14, *p* = 0.02), as was distance between patients’ residence to DTCs (OR = 1.03, CI: 1.01–1.05, *p* < 0.001), while non TB/HIV coinfected patients were 55% less likely to experience anxiety than those coinfected (OR = 0.45, CI: 0.29–0.71, *p* = 0.02; Table 6).

For a one unit increase in the household size, the odds of reporting stress were 1.08 times greater (OR = 1.08, CI: 1.03, 1.14, *p* = 0.002). Non-co-infected TB patients were 55% less likely to be stressed than co-infected ones (OR = 0.45, CI: 0.29–0.71, *p* < 0.001). For an increase of one unit in the distance from patients’ residence to DTCs, the odds of anxiety were 1.03 times greater (OR = 1.03, CI: 1.01–1.05, *p* < 0.001). Similarly, patients whose frequency of appointments for drug supply was between 15 and 30 days were 65% less likely to be stressed than those whose frequency of appointments for drug supply was ≤14 days (OR = 0.35, CI: 0.14–0.75, *p* = 0.011). Likewise, patients whose frequency of appointments for drug supply was >30 days were 99.94% less likely to experience stress than those whose frequency of appointments for drug supply was ≤14 days (OR = 0.06, CI: 0.00, 0.30, *p* = 0.007; (Table 7).

### 3.3. Qualitative Findings

In total, 36 TB patients, including 15 women, were interviewed about their perceptions regarding the COVID-19 pandemic. Thematic analysis was used to identify the following three predominate themes, and six sub-themes that emerged from the quantitative data.

**Theme** **1.**
*Social and psychological aspect.*


**Sub-theme.** 
*TB patients’ perception about the stigma related to COVID-19.*


Different opinions were expressed regarding the social and psychological impact of COVID-19. A commonly expressed concern was related to stigma, with the majority of participants accepting that stigma is commonly experienced by TB patients, and who also felt that the COVID-19 pandemic has increased the amount of stigma towards people with symptoms that may be suggestive of COVID-19. As one participant noted, “*Once you cough or sneeze, people distance themselves from you; then you feel bad… they think you have COVID-19 if you cough once or twice. For those who do not know that you have TB, they are going to think that you have COVID-19 because we do not overthink TB now*”. One participant shared how some people, without any knowledge on this person’s health status, do not even hesitate to walk away from them with a contemptuous gaze; other TB patients who were interviewed revealed how this practice has become more obvious during the COVID-19 pandemic, with people assuming symptoms such as coughs are an indication of COVID-19, rather than TB, demonstrated by the following statement: *“… ‘It’s so clear to me because my husband doesn’t allow me to come near him since he learned that COVID-19 also causes coughs. Usually, I did everything with him. But now, everything has changed, he no longer approaches me like before coronavirus pandemic, we no longer talk too much, and even now we eat separately, you see…*”.

**Sub-theme.** 
*Description of the health status of TB patients during COVID-19 pandemic.*


The majority of participants (*n* = 25) believed that their overall health conditions were better before the start of the COVID-19 pandemic. Some participants reported little hope for a full recovery of their health, due to the difficulties they faced in accessing effective TB treatment during the pandemic and the increasingly difficult living conditions under COVID-19, such as challenges with accessing food, demonstrated by the following statement: “*Mentally, I feel like garbage; all I’m worried about right now is how to get my health back. I don’t feel well right now, you see that by yourself! Due to food problems I barely earn, it’s just filling my stomach. Currently, it’s not okay, and sometimes I don’t have transport to get to the centre or sometimes even if you leave you are told that the centre is out of medication. I have lost a lot of weight at the moment. I lost much hope despite the encouragement of the doctors*”.

**Theme** **2.**
*Economic aspects.*


**Sub-theme.** 
*Financial situation of TB patients during the COVID-19 pandemic.*


Respondants highlighted how the cessation of certain activities during the COVID-19 pandemic directly and/or indirectly affected their financial conditions, which created challenges for accessing TB services and maintaining their treatment as this participant, who earns money to support their family through artistic activities, noted in the following statement: *“…. Financially it [the situation during COVID-19] is not good. There are no events, and as I am an artist, this is where I earn money to feed my family and pay for transportation to the* [treatment] *centre. But as there is currently no event, even finding transport to get to the treatment centre is a problem because transport is expensive, sometimes we do not come on the date that the doctors tell us to come, because we don’t have transportation. You see, it’s not easy”.*

**Sub-theme.** 
*Diet of TB patients during the COVID-19 pandemic.*


TB patients revealed during their interviews that their health condition had deteriorated significantly during COVID-19 due to the lack of an appropriate diet. They noted how they were no longer able to comply with the diet recommended by their doctors, as this participant noted in the following statement: “*I cannot eat all of the foods my doctor recommends. I eat with difficulty, and for example, with this difficult situation, I only eat rice. I have diabetes, and my doctor has advised me to limit my intake a bit. But I have to consume it because that’s what I earn”.*

**Theme** **3.**
*Perceptions on government actions during the COVID-19 pandemic.*


**Sub-theme.** 
*Restrictive measures.*


According to our participants, the Guinean population’s life depends on their daily economic activities; therefore, the restrictive measures imposed by the Government to limit the spread of COVID-19 were largely regarded as drastic and thought to cause more harm than good, as shown by the following statement: “*These measures of the state are salutary, but which is not without consequences for some, you knew it as well as me sir… these measures make life difficult by blocking the activities of people, and you know here what we see, that’s what we eat. Guineans eat from day to day, so these measures increase their suffering*”.

**Sub-theme.** 
*Mitigation measures.*


Unanimously, respondents believed that the Government should financially help its people, especially those who were sick, poor, children or elderly, demonstrated by the following statement: “*I thank the state for what is already done the price of fuel at the pump, this leads to a reduction in the prices of necessities (rice, sugar …), and, therefore facilitates access to food by the population. But some do not benefit from it and others do not even know about it. But when the price drops, everyone will feel this impact on their living condition*”. Likewise, some believed that special attention should be paid to TB patients given the additional burdens created by their illness, as shown by the following statement: “*Yes, the state should help its population, especially patients like us tuberculosis patients who are rejected by our bosses for fear of being infected ……”.*

## 4. Discussion

The advent of the COVID-19 pandemic has caused a noticeable imbalance in the smooth running of our society. Healthy and sick individuals feel this impact differently. Regardless of any external disturbances, TB patients constitute a vulnerable group. This study showed significant additional discomfort for these TB patients. Even though our study coincided with the onset of relaxation of the restrictive measures, we found a significant proportion of patients reporting pain or discomfort, and anxiety or depression. Our results are consistent with a recent evaluation of the stress response during COVID-19, in which a significant proportion of patients struggled to cope with stress [24]. Apart from the situation induced by the COVID-19 pandemic, psychological problems are quite recurrent in TB patients and should be the subject of special attention [13,14]. One of the reasons for this psychosocial state of patients stems from the stigma experienced by TB patients during COVID-19, given the similarity between these two diseases [25,26]. In addition, in Guinea, the implementation of restrictive measures limited the population’s movements and led to the closure of many businesses, which further reduced many patients’ support sources, worsening financial problems and exacerbating stress. Our findings underscore the need for comprehensive psycho-social support to TB patients during times of public health emergencies to protect them against further vulnerability.

In our study, we also sought to understand the factors that explained the different quality of life scales (EQ 5D 3L). Each item (pain or discomfort, usual activities and anxiety or depression) of EQ-5D-3L made it possible to identify the factors associated with them. Socio-demographically, it emerges that age, the number of people in the household, and the distance between patients’ homes and the care site explain the deterioration in life quality. Among the factors that degrade one’s quality of life, we have, unsurprisingly, the elderly, who presented more difficulties in carrying out daily activities than children. This category of patients is potentially subject to various pathologies and dysfunctions. Due to reduced immunity, their fragility exposes them to more significant risks, especially side effects [14,27,28] and malnutrition. Unlike young people, TB in the elderly poses a real diagnostic and management challenge [26,28,29,30]. However, we must be aware that the management of diseases in the elderly both in the context and outside of the COVID-19 pandemic presents challenges in the Guinean context, where suitable places and services to take care of vulnerable people are lacking. As such, whenever we are faced with stressful situations, these elderly people are the first to feel the impact. The number of people in households has also emerged as a factor associated with mobility problems, pain or discomfort, anxiety and stress. This situation could be explained by patients’ economic situations, who lack adequate care and, as a result, are exposed to these various traumatic events. The socioeconomic conditions are also manifested by the fact that the patients who were far from care had a degraded HRQoL. Indeed, not often having the means to travel regularly to health centres increases considerably the risk of having a degraded HRQoL.

We found that patients clinically diagnosed, coinfected or with extra-pulmonary or drug-resistant TB were dominant factors associated with a deterioration in HRQoL. Previously treated patients had a lower risk of poor quality of life compared to patients treated more recently. Newly diagnosed patients struggle to adapt to the disease’s new life, especially in a particular pandemic context. HRQoL strongly depends on clinical forms of tuberculosis [5], which depend on apparent clinical and subclinical manifestations. The approximate duration of the anti-tuberculosis drug procurement remains the primary factor identified in this study. Patients expressed many difficulties related to travel and the financial problems involved. We have observed that the shorter time for drug procurement, the more the patients’ HRQoL deteriorated. Thus, the close approximation of supply time increases the cost and the fear and discomfort when travelling. Tuberculosis patients in the African context often face catastrophic health costs related to their living conditions [31]. Therefore, any gait that requires frequent movement affects the HRQoL.

Beyond the associated factors revealed by the quantitative method, the other recurring element expressed by patients in the qualitative part is malnourishment. The nutritional support program only benefits multidrug-resistant patients in Guinea, leaving the rest of the TB patients neglected. This is not without consequences for tuberculosis patients, and for this reason, the substantial cessation of activities has impacted the possibility for these tuberculosis patients to meet their nutritional needs.

Our study has the advantage of combining two complementary approaches. However, the study design did not allow us to fit all the quantitative questions into the qualitative scheme. This made the comparison of the two approaches difficult. It should also be noted that the study’s implementation coincided with the start of the relaxation of restrictive measures. This implies that the resulting image would be even worse at the onset of COVID-19 in Guinea. In any case, a comparison of the different periods would have been useful. We were unable to fully explore the real economic impact (associated costs) on patients’ lives and data on the patients’ direct experience with COVID-19. The simplification of the study design, the need for information within a short time, social distancing concerns, and other barrier measures limited our ability to go into depth in the assessment. Although we were unable to survey tuberculosis patients outside the capital, our sample remains representative of most tuberculosis patients in the Guinean context, because the Capital Conakry takes care of 60% of tuberculosis patients. However, we acknowledge that by recruiting TB patients only from DTCs, we may have missed those unable to attend for reasons such as fear of or exposure to COVID-19, which may have exacerbated the quality of life challenges beyond those that we have reported here. Lastly, we acknowledge that the lack of a comparator for HRQoL among TB patients during the pre-COVID-19 period, or among non-TB patients during the pandemic, limits our ability to draw causal links about the impact of COVID-19 on the HRQoL among TB patients in Guinea; however, we believe the data presented reflect both the existing literature on the impact of COVID-19 on TB patients and the experience of the authors, and is a useful starting point for further exploration to ameliorate the additional challenges caused by the COVID-19 pandemic.

## 5. Conclusions

This study confirmed the common problems that influence tuberculosis patients’ quality of life and how such challenges are exacerbated by the COVID-19 pandemic in Guinea. A mitigation plan that considers patients with particular clinical characteristics, nutritional issues, and strategies to reduce tuberculosis patients’ movement for accessing TB care during the COVID-19 pandemic is urgently needed. Finally, this plan must necessarily include a psychosocial care component.

## Figures and Tables

**Figure 1 tropicalmed-07-00224-f001:**
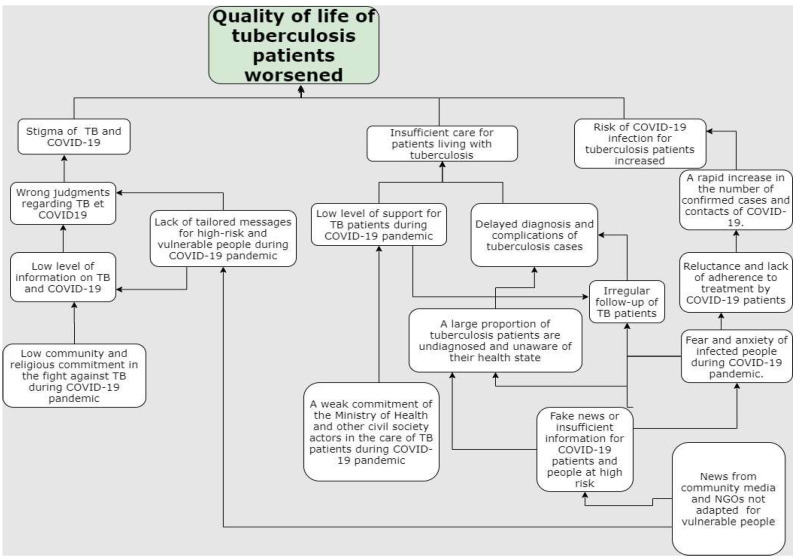
Hypothesized problem tree of HQoL of TB patients during the COVID-19 pandemic in Guinea.

**Table 1 tropicalmed-07-00224-t001:** Socio-demographic characteristics of the study sample (*n* = 439).

	Variable	*n* (%)
Patient age group		
	Children (<18 years)	22 (5.0%)
	Young (19–39 years)	258 (59%)
	Adults (40–59 years)	125 (28%)
	Elderly (>59 years)	34 (7.7%)
Gender		
	Male	269 (61%)
	Female	170 (39%)
Marital status		
	Single	269 (61%)
	Married	170 (39%)
Residence		
	Matoto	168 (38%)
	Ratoma	97 (22%)
	Matam	95 (22%)
	Dixinn	15 (3.4%)
	Kaloum	32 (7.3%)
	Outside Conakry	32 (7.3%)
Level of education		
	None/informal education only	133 (30%)
	Primary level (6 years)	86 (20%)
	Secondary level (10–13) years)	158 (36%)
	High school level (≥10 years)	51 (12%)
	Tertiary/university level (≥14)	11 (2.5%)
Occupation type		
	None/unemployed	133 (29.8%)
	Private employment	40 (9.1%)
	Civil servant/public sector	18 (4.1%)
	Freelance/self employed	248 (56%)
Average number of household members (range)		6 (1–30)
Mean distance between the patient’s home and the treatment site (in kilometers)		6.96 (2–100)

**Table 2 tropicalmed-07-00224-t002:** Clinical characteristics of the study sample (*n* = 439).

	Variable	*n* (%)
Tuberculosis type		
	Drug sensitive	394 (90%)
	Drug resistant	45 (10%)
TB disease localization		
	Pulmonary	354 (81%)
	Extrapulmonary	85 (19%)
TB treatment history		
	New case	390 (89%)
	Previously treated	49 (11%)
TB diagnosis type		
	Bacteriologically confirmed	379 (86%)
	Clinically diagnosed	60 (14%)
HIV status		
	Negative	317 (72%)
	Unknown	15 (3.4%)
	Positive	107 (24%)
	Started on ARV treatment	97 (90.7%)
Treatment initiation		
	Mean distance in kilometers between residence and TB treatment center (range)	3 (2, 8)
Frequency of appointments for drug supply (in days)		
	≤14 days	363 (83%)
	15–30 days	45 (10%)
	>30 days	31 (7.1%)
Duration of TB treatment (in days)		
	≤30 days	110 (25%)
	31–90 days	121 (28%)
	>90 days	208 (47%)

**Table 3 tropicalmed-07-00224-t003:** Type and severity of traumatic events reported by TB patients using theEQ-5D-3L scale and IES-R (*n* = 439).

Domain	Mean (SD ^1^)	Level of Impairment (%)		
		None	Moderate	Serious
Usual activities	1.31 (0.52)	72.2	21.1	2.7
Self-care	1.13 (0.38)	89.1	9.1	1.8
Pain and discomfort	1.49 (0.56)	53.5	43.5	3.0
Mobility	1.23 (0.45)	78.1	20.7	1.1
Anxiety and/or depression	1.43 (0.61)	63.1	30.5	6.4
IES-R	1.37 (0.65)	72.7	17.8	9.6

^1^ SD: standard deviation.

**Table 4 tropicalmed-07-00224-t004:** Multinomial logistic regression between difficulty with usual activities and sociodemographic and clinical characteristics.

Characteristic	AOR	95% CI	*p*-Value
Patient age group			
Children	—	—	
Young	1.34	0.46, 4.82	0.6
Adults	2.63	0.87, 9.72	0.11
Elderly	5.42	1.57, 22.2	0.011
Tuberculosis type			
Sensitive	—	—	
Resistant	27.7	6.02, 132	<0.001
Treatment history			
New case	—	—	
Already treated	0.02	0.00, 0.11	<0.001
TB diagnostic			
Bacteriological	—	—	
Clinical examination	2.31	1.20, 4.42	0.011
HIV status			
Positive	—	—	
Negative	0.52	0.32, 0.87	0.012
Distance between the patient’s home and the treatment site (in kilometers)	1.02	1.00, 1.04	0.062
Frequency of appointments for drug supply (in days)			
≤14	—	—	
15–30	0.50	0.19, 1.14	0.12
>30	0.07	0.01, 0.32	0.003
Duration of treatment (in days)			
≤30	—	—	
31–90	0.92	0.51, 1.67	0.8
>90	0.56	0.32, 0.98	0.043

AOR = adjusted odds ratio, CI = confidence interval.

**Table 5 tropicalmed-07-00224-t005:** Multinomial logistic regression pain/discomfort and sociodemographic and clinical characteristics.

Characteristic	AOR	95% CI	*p*-Value
Patient age group			
Children	—	—	
Young	0.67	0.27, 1.71	0.4
Adults	1.07	0.42, 2.83	0.9
Elderly	1.71	0.56, 5.29	0.3
Household member number	1.05	1.00, 1.11	0.045
Tuberculosis type			
Sensitive	—	—	
Resistant	1.83	0.96, 3.52	0.067
TB diagnostic			
Bacteriological	—	—	
Clinical examination	1.93	1.06, 3.54	0.031
RAV treatment			
Yes	—	—	
No	3.76	0.97, 15.0	0.056
Frequency of appointments for drug supply (in days)			
≤14	—	—	
15–30	0.34	0.16, 0.67	0.003
>30	0.50	0.21, 1.12	0.10

AOR = adjusted odds ratio, CI = confidence interval.

**Table 6 tropicalmed-07-00224-t006:** Multinomial logistic regression anxiety/depression and sociodemographic and clinical characteristics.

Characteristic	AOR	95% CI	*p*-Value
Household member number	1.08	1.03, 1.14	0.002
HIV status			
Positive	—	—	
Negative	0.45	0.29, 0.71	<0.001
Distance between the patient’s home and the treatment site (in kilometers)	1.03	1.01, 1.05	<0.001
Frequency of appointments for drug supply (in days)			
≤14	—	—	
15–30	0.35	0.14, 0.75	0.011
>30	0.06	0.00, 0.30	0.007

AOR = odds ratio, CI = confidence interval.

**Table 7 tropicalmed-07-00224-t007:** Multinomial logistic regression between stress and sociodemographic and clinical characteristics.

Characteristic	OR ^1^	95% CI ^1^	*p*-Value
Household member number	1.08	1.03, 1.14	0.002
HIV status			
Positive	—	—	
Negative	0.45	0.29, 0.71	<0.001
Distance between the patient’s home and the treatment site (in kilometers)	1.03	1.01, 1.05	<0.001
Frequency of appointments for drug supply (in days)			
≤14	—	—	
15–30	0.35	0.14, 0.75	0.011
>30	0.06	0.00, 0.30	0.007

^1^ OR = odds ratio, CI = confidence interval.

## Data Availability

Requests for original datasets used in this manuscript can be directed to the corresponding author.

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
