# Peer review of "Health-Related Quality of Life of Tuberculosis Patients during the COVID-19 Pandemic in Conakry, Guinea: A Mixed Methods Study"

_tropicalmed, 2022, doi:10.3390/tropicalmed7090224_

Round 1
Reviewer 1 Report
The authors based their study on the premise that the COVID-19 pandemic impacted all facets of life and exacerbated the challenges of people with tuberculosis (TB). To assess the health-related quality of life (HRQoL) of TB patients in Guinea during the COVID-19 pandemic, they conducted a study using two validated psychometric tools on a population of TB patients from three centers in Conakry. From 439 participants, the results showed that a higher number of household members and a greater distance between their residence and the health center negatively affect HRQoL. They also noted an increase in nutritional and financial concerns, exacerbated during the COVID-19 pandemic, and a belief that the government support plan was insufficient. Based on their findings, the authors conclude that specific support plans should be implemented for TB patients, including nutritional and psychological support, particularly for those with travel restrictions, reduced employment opportunities, and financial needs.
Although their findings contribute to the field, the authors must properly address the following minor concerns before publishing the study.
1. Abstract: clear and fluent; however, reviewing the construction of some statements will reduce possible confusion in readers unfamiliar with the topic (use simple words and reader-based writing).
2. Introduction: It contains the scientific foundations that support the study. Also, use reader-based writing to avoid losing the study's meaning and impact.
3. Materials and Methods. Subsection: Study Framework Analysis, Page 3, Lines 1-11. Please correct that paragraph as it reads confusing. Also, reduce the overuse of the word "hypothesis" and its derivatives in the first lines. Also, Figure 1 could be self-explanatory (increase the font size and use all available margins).
4. Results. Pages 6 and 7. Table 1 is overloaded. Please consider splitting it into two tables, one containing the socio-demographic characteristics and the other the clinical characteristics of the study population.
Given the above, the manuscript is endorsed for publication in "Tropical Medicine and Infectious Disease" once minor concerns are addressed.
Author Response
Dear Editor,
We appreciate you and the reviewers for your precious time in reviewing our paper and providing valuable comments. It was your valuable and insightful comments that led to possible improvements in the current version. The authors have carefully considered the comments and tried our best to address every one of them. We hope the manuscript after careful revisions meet your high standards. The authors welcome further constructive comments if any.
Below we have provided a detailed response to each of the points raised by the reviewers.
On behalf of the authors
Almamy Amara TOURE, MD, MSPH.
National Centre for Training and Research in Rural Health of Mafèrinyah.
Head of Monitoring and evaluation unit.
Response to Reviewer 1 Comments
Reviewer #1 Minor changes
Open Review
(x) I would not like to sign my review report
( ) I would like to sign my review report
English language and style
( ) Extensive editing of English language and style required
( ) Moderate English changes required
(x) English language and style are fine/minor spell check required
( ) I don't feel qualified to judge about the English language and style
|
Yes |
Can be improved |
Must be improved |
Not applicable |
|
|
Does the introduction provide sufficient background and include all relevant references? |
( ) |
(x) |
( ) |
( ) |
|
Are all the cited references relevant to the research? |
(x) |
( ) |
( ) |
( ) |
|
Is the research design appropriate? |
(x) |
( ) |
( ) |
( ) |
|
Are the methods adequately described? |
( ) |
(x) |
( ) |
( ) |
|
Are the results clearly presented? |
( ) |
(x) |
( ) |
( ) |
|
Are the conclusions supported by the results? |
(x) |
( ) |
( ) |
( ) |
Comments and Suggestions for Authors
The authors based their study on the premise that the COVID-19 pandemic impacted all facets of life and exacerbated the challenges of people with tuberculosis (TB). To assess the health-related quality of life (HRQoL) of TB patients in Guinea during the COVID-19 pandemic, they conducted a study using two validated psychometric tools on a population of TB patients from three centers in Conakry. From 439 participants, the results showed that a higher number of household members and a greater distance between their residence and the health center negatively affect HRQoL. They also noted an increase in nutritional and financial concerns, exacerbated during the COVID-19 pandemic, and a belief that the government support plan was insufficient. Based on their findings, the authors conclude that specific support plans should be implemented for TB patients, including nutritional and psychological support, particularly for those with travel restrictions, reduced employment opportunities, and financial needs.
Although their findings contribute to the field, the authors must properly address the following minor concerns before publishing the study.
- Abstract: clear and fluent; however, reviewing the construction of some statements will reduce possible confusion in readers unfamiliar with the topic (use simple words and reader-based writing).
Response
We have edited the abstract section to be more in line with this recommendation. Please see our changes made throughout the abstract.
- Introduction: It contains the scientific foundations that support the study. Also, use reader-based writing to avoid losing the study's meaning and impact.
Response
We have edited the introduction section to be more in line with this recommendation. Please see our changes made throughout the introduction.
- Materials and Methods. Subsection: Study Framework Analysis, Page 3, Lines 1-11. Please correct that paragraph as it reads confusing. Also, reduce the overuse of the word "hypothesis" and its derivatives in the first lines. Also, Figure 1 could be self-explanatory (increase the font size and use all available margins).
Response
We have made edits to the study framework analysis section Page 3 (lines 15 – 32)
- Pages 6 and 7. Table 1 is overloaded. Please consider splitting it into two tables, one containing the socio-demographic characteristics and the other the clinical characteristics of the study population.
Response
We agree with this suggestion and have modified Table 1 into Table 1 and 2.
Given the above, the manuscript is endorsed for publication in "Tropical Medicine and Infectious Disease" once minor concerns are addressed.
Submission Date
21 July 2022
Date of this review
07 Aug 2022 04:47:13
Reviewer 2 Report
Health-Related Quality of Life of Tuberculosis Patients During the COVID-19 Pandemic in Conakry, Guinea: A Mixed Methods Study
The study of Touré et al. underlines the dire need to address the difficulties that TB-infected people face in Guinea in the wake of the COVID-19 pandemic.
This strength of this study is that it points out the key factors (stigmatization, travel difficulties, financial problems, malnourishment and the stress resulting from these problems) that cause the most difficulty to the patients in Guinea in their adherence to the antitubercular treatment. Since non-adherence can lead to treatment failure, relapse and the development of drug-resistant TB strains, finding these key points is essential in aiding the government and appropriate agencies to develop a response plan that ameliorates the situation. As Guinea entered the WHO’s list of the 30 high burden country for HIV-associated TB in 2021 (Global tuberculosis report 2021 ISBN 978-92-4-003702-1), it is important to consider the need of this population also.
Since very little is known about the HRQoL among TB patients in Guinea before the COVID19 pandemic, it is hard to evaluate the impact of COVID-19 on TB patients' HRQoL. Nevertheless, this study provides important data about the current situation of TB-patients and can serve as a starting point for future studies.
My questions and notes for the Authors:
The authors mention that the mean distance from patients' homes to the DTC was 3 km, and also that patients that were far from care, had a degraded HRQoL. What was the range and distribution of the distance between these 2 places among the participants?
Did all the participants had active TB disease? If not, what was the LTBI and Active TB patient ratio and how did this affect the different aspects of HRQoL measured (e.g. the perception of the stigma associated with coughing)?
One participant mentions that they are diabetic and have problems regarding their diet. Smoking, alcohol use, diabetes and pregnancy can also affect the HRQoL during TB disease and antitubercular treatment, and therefore, adherence to treatment. Was information collected from the participants regarding these variables?
Are “length of time for drug procurement (days)” and “duration of the appointment for drug supplies (days)” are the same variables? Please explain their meaning in the text. The authors mention that patients required to attend daily at the TB clinics for directly observed treatment (page 1, row 46 - page 2 row 1), but the results (Tables 3-5) and the discussion (page13, rows 5-7) suggest to the reader that the patients are given a supply of drugs to take home and the size of this supply determines how often they need to travel to the clinics. Which one is correct?
The duration of treatment is mentioned in Table 3 and in several occasions in the text. Please include it’s variables and n (%) values of the study sample regarding this charasteristic in Table 1.
There is incongruence between the text on Page 10, rows 5-9 and Table 6: Values missing from the “distance” row of the table and values in the “Duration of the appointment for drug supplies” rows are listed as values for “distance between site and residence” and “extrapulmonary TB” in the text. “Duration of treatment” values, that are mentioned in the text, are missing from the table.
Page 12 Row 19: Reference 23 does not support the statement about COVID-19 and/or stress in TB patients.
Please check text for typos, e.g. page 8 rows 2-12.
Author Response
Dear Editor,
We appreciate you and the reviewers for your precious time in reviewing our paper and providing valuable comments. It was your valuable and insightful comments that led to possible improvements in the current version. The authors have carefully considered the comments and tried our best to address every one of them. We hope the manuscript after careful revisions meet your high standards. The authors welcome further constructive comments if any.
Below we have provided a detailed response to each of the points raised by the reviewers.
On behalf of the authors
Almamy Amara TOURE, MD, MSPH.
National Centre for Training and Research in Rural Health of Mafèrinyah.
Head of Monitoring and evaluation unit.
Response to Reviewer 2 Comments
Open Review
(x) I would not like to sign my review report
( ) I would like to sign my review report
English language and style
( ) Extensive editing of English language and style required
(x) Moderate English changes required
( ) English language and style are fine/minor spell check required
( ) I don't feel qualified to judge about the English language and style
|
Yes |
Can be improved |
Must be improved |
Not applicable |
|
|
Does the introduction provide sufficient background and include all relevant references? |
(x) |
( ) |
( ) |
( ) |
|
Are all the cited references relevant to the research? |
( ) |
(x) |
( ) |
( ) |
|
Is the research design appropriate? |
( ) |
(x) |
( ) |
( ) |
|
Are the methods adequately described? |
(x) |
( ) |
( ) |
( ) |
|
Are the results clearly presented? |
(x) |
( ) |
( ) |
( ) |
|
Are the conclusions supported by the results? |
( ) |
( ) |
( ) |
( ) |
Comments and Suggestions for Authors
Health-Related Quality of Life of Tuberculosis Patients During the COVID-19 Pandemic in Conakry, Guinea: A Mixed Methods Study
The study of Touré et al. underlines the dire need to address the difficulties that TB-infected people face in Guinea in the wake of the COVID-19 pandemic.
This strength of this study is that it points out the key factors (stigmatization, travel difficulties, financial problems, malnourishment and the stress resulting from these problems) that cause the most difficulty to the patients in Guinea in their adherence to the antitubercular treatment. Since non-adherence can lead to treatment failure, relapse and the development of drug-resistant TB strains, finding these key points is essential in aiding the government and appropriate agencies to develop a response plan that ameliorates the situation. As Guinea entered the WHO’s list of the 30 high burden country for HIV-associated TB in 2021 (Global tuberculosis report 2021 ISBN 978-92-4-003702-1), it is important to consider the need of this population also.
Since very little is known about the HRQoL among TB patients in Guinea before the COVID19 pandemic, it is hard to evaluate the impact of COVID-19 on TB patients' HRQoL. Nevertheless, this study provides important data about the current situation of TB-patients and can serve as a starting point for future studies.
My questions and notes for the Authors:
The authors mention that the mean distance from patients' homes to the DTC was 3 km, and also that patients that were far from care, had a degraded HRQoL. What was the range and distribution of the distance between these 2 places among the participants?
Response
Thank for this question. We have corrected the mean distance from patients ‘homes to the DTC and given the range. Please see P6 from 36 to 38.
Did all the participants had active TB disease? If not, what was the LTBI and Active TB patient ratio and how did this affect the different aspects of HRQoL measured (e.g. the perception of the stigma associated with coughing)?
Response
Thank you for your observation. Although you have raised an important aspect, we did not seek the LTBI. As you can notice, we only included active TB patients.
One participant mentions that they are diabetic and have problems regarding their diet. Smoking, alcohol use, diabetes and pregnancy can also affect the HRQoL during TB disease and antitubercular treatment, and therefore, adherence to treatment. Was information collected from the participants regarding these variables?
Response
As for comorbidity like diabetes, alcohol we acknowledge their value for more explanations to the HRQoL, but we did not collect them and consider to put this part in the limitations of the study. Please see P15 line 13.
Are “length of time for drug procurement (days)” and “duration of the appointment for drug supplies (days)” are the same variables? Please explain their meaning in the text. The authors mention that patients required to attend daily at the TB clinics for directly observed treatment (page 1, row 46 - page 2 row 1), but the results (Tables 3-5) and the discussion (page13, rows 5-7) suggest to the reader that the patients are given a supply of drugs to take home and the size of this supply determines how often they need to travel to the clinics. Which one is correct?
Response
Thanks for pointing that out. For “length of time for drug procurement (days)” and “duration of the appointment for drug supplies (days)”, we harmonized in the manuscript with “the frequency of appointments for drug supply (in days)” as suggested by one of the reviewers.
Indeed, the classic protocol for managing TB patients is based on the principle of directly observed treatment. Still, due to the constraints linked to this strategy, the drugs of some patients are given to them for a period. In addition, we must add that COVID-19 has reinforced this situation of providing the drugs to patients due to limitations in movement.
The duration of treatment is mentioned in Table 3 and in several occasions in the text. Please include it’s variables and n (%) values of the study sample regarding this charasteristic in Table 1.
Response
We have included the duration of treatment in Table 2 as table 1 splitting suggested by one of the reviewers. Please P8 Table 2. There is incongruence between the text on Page 10, rows 5-9 and Table 6: Values missing from the “distance” row of the table and values in the “Duration of the appointment for drug supplies” rows are listed as values for “distance between site and residence” and “extrapulmonary TB” in the text. “Duration of treatment” values, that are mentioned in the text, are missing from the table.
Response
Thank you for your observation. We have updated the table to be congruent with the text. Please see P11 and P12.
Page 12 Row 19: Reference 23 does not support the statement about COVID-19 and/or stress in TB patients.
Response
We have carefully reviewed and correct the reference. Please see 14 and P17.
Please check text for typos, e.g. page 8 rows 2-12.
Response
We have checked for typos through the manuscript.
Submission Date
21 July 2022
Date of this review
15 Aug 2022 20:02:12
Reviewer 3 Report
This is a useful study that provides some preliminary information on the problems faced by patients with TB during the covid 19 pandemic. However, the study does not enable identification of the potential impacts of the covid 19 pandemic, as little information is provided on the status of the pandemic or pandemic responses during the period covered by the study. This could be addressed by providing more information on the pandemic context and responses, and / or some caution in the interpretation of the results in terms of pandemic impacts. There is also a need for some editing of language, and clarification of some elements of the text and tables.
More detailed comments follow
Abstract: clarify the timing of the study in relation to the stage of the covid pandemic in Guinea
Introduction
P 1 line 44 TB notification among children – what is the issue here ? eg delays or reluctance to notify
P 2 line 9 provide further information on the extent, timing and responses to Covid 19 in Guinea and how these might impact patients with TB
P 2 line 31-32 ‘compared to before’ – is there any information on HRQoL of TB patients in Guinea prior to this study ? if so please summarise.
Materials and methods
P 3 line 2 ‘reductions in reported HRQoL’ – this could only be determined if there were previous measures of HRQoL. In the absence of previous measurements, the study can only examine the current HRQoL and identify any factors related to the covid-19 pandemic
How was the hypothesized problem tree developed ?
P3 line 22: what proportion of the total DTCs are included ? how were the study DTCs selected ? how representative is this sample population ? how do the dates of the study relate to the evolution of the covid 19 pandemic ?
P 4 data collected. The data doesn’t appear to include any data on the patient’s direct experience with covid 19 – for example whether self or family members have had suspected or diagnosed covid 19 ; whether they have had to isolate because of contact with covid 19 etc
Some of these factors may be covered in the qualitative interviews although the questions appear to focus more on attitudes to covid 19.
P 4 sampling and recruitment. Consider whether recruitment from those attending DTCs may have excluded those unable to attend due to covid 19 contact, or concerns about exposure to covid 19. The potential impact of the sample should be considered in the discussion on the results.
P 5 line 34 government’s response to covid 19. It would be helpful to include a description of the evolution of the covid 19 pandemic and responses from the government, particularly in regards to those that might impact on TB patients, in the description of the setting.
Results
P 5 Line 45 This is the first information on the HIV epidemic and the fact that a significant proportion of the patients had both HIV and TB. Information on the HIV situation and its relationship with TB could be provided in the settings, and needs to be considered as a potential complication in terms of HRQoL in the problem tree.
P 8 Multivariate analysis. It is unclear what is the outcome measure with which the factors are associated in Table 3: re-word the title to state ‘difficulty with usual activities’ rather than just usual activities, if this is the outcome measure.
Some of the factors require clearer description. For example does ‘age’ refer to the age of the patient or to the presence of people of these ages in the household ?
P 9 line 3 and Table 4. The association with household size is unclear – as the number of household members or reference value is not mentioned. It is unclear what is meant by ‘time for drug procurement’ – which appears to be the term ‘duration of appointment for drug supplies’ in the table. This might be clearer by using wording such as ‘frequency of appointments for collecting drug supply’.
P 9 line 8 provides a somewhat clearer statement of what is meant by ‘household size’ in relation to association with anxiety, although the clinical significance is unclear.
P 10 Table 6 does not provide any figures for the factor ‘distance between site and residence’ although the text (line 5) refers to an AOR of 1.03, which, in the table 6, is provided for a duration of appointment of 15-30 days. The figure for duration of treatment of > 30 days is not provided in Table 6 although quoted in the text at line 7; nor is the data for extrapulmonary TB.
Qualitative findings
This section provides a clear description of some interesting findings. However, more information on the restrictions and measures introduced to control covid 19 in the settings section would assist the reader in understanding how these might impact on patients with TB.
Discussion
The reports of pain, discomfort, anxiety and depression by TB patients in the questionnaire are not compared to reported levels prior to the covid 19 pandemic, or to persons who do not have TB, and thus it is not possible to determine how these might have changed as a result of covid 19, or how these might compare with healthy individuals. The wording in the discussion needs to reflect some caution in interpretation.
The information from the qualitative interviews does enable some assessment of the impact of covid 19, although some of the comments (for example those related to livelihood) may well also apply to persons without TB.
Caution is needed in the interpretation of the quality of life scale results, as the survey does not enable an analysis of the causative factors that might contribute to the differences among scales for different factors. For example, it is unclear whether the higher level of problems with daily activities or pain and discomfort for the elderly refer to problems of the elderly or problems of the household in managing elderly members. The discussion seems to confuse issues of the clinical management of elderly TB patients, together with issues of managing elderly household members.
The comments on p 13 regarding caution in interpretation of the findings are important and helpful.
Author Response
Dear Editor,
We appreciate you and the reviewers for your precious time in reviewing our paper and providing valuable comments. It was your valuable and insightful comments that led to possible improvements in the current version. The authors have carefully considered the comments and tried our best to address every one of them. We hope the manuscript after careful revisions meet your high standards. The authors welcome further constructive comments if any.
Below we have provided a detailed response to each of the points raised by the reviewers.
On behalf of the authors
Almamy Amara TOURE, MD, MSPH.
National Centre for Training and Research in Rural Health of Mafèrinyah.
Head of Monitoring and evaluation unit.
Response to Reviewer 3 Comments
Reviewer #3 Major changes
Open Review
( ) I would not like to sign my review report
(x) I would like to sign my review report
English language and style
( ) Extensive editing of English language and style required
( ) Moderate English changes required
(x) English language and style are fine/minor spell check required
( ) I don't feel qualified to judge about the English language and style
|
Yes |
Can be improved |
Must be improved |
Not applicable |
|
|
Does the introduction provide sufficient background and include all relevant references? |
( ) |
( ) |
(x) |
( ) |
|
Are all the cited references relevant to the research? |
(x) |
( ) |
( ) |
( ) |
|
Is the research design appropriate? |
( ) |
(x) |
( ) |
( ) |
|
Are the methods adequately described? |
( ) |
(x) |
( ) |
( ) |
|
Are the results clearly presented? |
( ) |
( ) |
(x) |
( ) |
|
Are the conclusions supported by the results? |
( ) |
( ) |
(x) |
( ) |
Comments and Suggestions for Authors
This is a useful study that provides some preliminary information on the problems faced by patients with TB during the covid 19 pandemic. However, the study does not enable identification of the potential impacts of the covid 19 pandemic, as little information is provided on the status of the pandemic or pandemic responses during the period covered by the study. This could be addressed by providing more information on the pandemic context and responses, and / or some caution in the interpretation of the results in terms of pandemic impacts. There is also a need for some editing of language, and clarification of some elements of the text and tables.
More detailed comments follow
Author response:
Thank you for your comments, which we have done our best to integrate into the updated version of the manuscript. Below we have provided a detailed response to each of the points raised by the reviewers:
Abstract: clarify the timing of the study in relation to the stage of the covid pandemic in Guinea
Response: We have clarified the timing of the study in relation to the stage of the covid pandemic in Guinea as the” first wave”. Please see in the abstract section from line 21 to 22.
Introduction
P 1 line 44 TB notification among children – what is the issue here? eg delays or reluctance to notify
Response: We have edited this section to clarify the issue as “under notification” of TB among children. Please see P1 line 44.
P 2 line 9 provide further information on the extent, timing and responses to Covid 19 in Guinea and how these might impact patients with TB
Response: We have added additional details in the manuscript to address this comment. Please see p2 from line 10 to 30.
P 2 line 31-32 ‘compared to before’ – is there any information on HRQoL of TB patients in Guinea prior to this study? if so please summarise.
Response
We note that there is no current information on the quality of life of TB patients before COVID-19 in Guinea specifically, but we have based our hypothesis on the literature which suggests indicates that an association between worsening quality of life and TB and COVID-19, respectively.
Materials and methods.
P 3 line 2 ‘reductions in reported HRQoL’ – this could only be determined if there were previous measures of HRQoL. In the absence of previous measurements, the study can only examine the current HRQoL and identify any factors related to the covid-19 pandemic
Response
We agree with your observations. We wanted to indicate that even without COVID-19, the quality of life of TB patients is impaired in the absence of any intervention. For example, a systematic review of the quality of life of TB patients based on EQ-5D indicates a decrease in it and its improvement with TB treatment. Therefore, we have rephrased this part by stating, "given the deterioration in the quality of life of TB patients even outside the context of COVID-19, we hypothesize a risk of deterioration". You can see changes on p3 from lines 15 to 18.
How was the hypothesized problem tree developed?
Response
The hypothesized problem tree was developed through discussion between the key actors of Guinea's national TB program. We have edited the text to clarify this; please see p3 from lines 18 to 19.
P3 line 22: what proportion of the total DTCs are included? how were the study DTCs selected? how representative is this sample population? how do the dates of the study relate to the evolution of the covid 19 pandemic?
Response.
We have added additional text to address this question. Please refer to lines 14 to 16 on page 4.
P 4 data collected. The data doesn’t appear to include any data on the patient’s direct experience with covid 19 – for example whether self or family members have had suspected or diagnosed covid 19; whether they have had to isolate because of contact with covid 19 etc
Response
Unfortunately, we did not include any of these data as part of the study, however we have added additional text to the discussion to mention this as a study limitation. Please see p15 from lines 10 to 11. 29.
P 4 sampling and recruitment. Consider whether recruitment from those attending DTCs may have excluded those unable to attend due to covid 19 contact, or concerns about exposure to covid 19. The potential impact of the sample should be considered in the discussion on the results.
Response
We note that our sample was limited to those attending the CDT but acknowledge that this may have created a bias in the results by excluding those who were unable to attend due to the reasons mentioned. We have added some additional text to include this point in the discussion (lines 15 – 19, page 15).
P 5 line 34 government’s response to covid 19. It would be helpful to include a description of the evolution of the covid 19 pandemic and responses from the government, particularly in regards to those that might impact on TB patients, in the description of the setting.
Response
We have added additional information on the national response to COVID-19 in the background. You can see p2 from lines 12 to 26.
Results
P 5 Line 45 This is the first information on the HIV epidemic and the fact that a significant proportion of the patients had both HIV and TB. Information on the HIV situation and its relationship with TB could be provided in the settings, and needs to be considered as a potential complication in terms of HRQoL in the problem tree.
Response
Thank you for this recommendation, however, as the problem tree was developed a priori, we do not feel it is appropriate to amend the tree based on the results. Furthermore, as the majority of the HIV/TB coinfected patients were successfully being treated with ARVs, we feel that the contribution of HIV infection to the worsening of HRQoL, relative to the COVID-19 pandemic and TB, would be minor. While we appreciate this recommendation, respectfully, we do not feel that changes are required.
P 8 Multivariate analysis. It is unclear what is the outcome measure with which the factors are associated in Table 3: re-word the title to state ‘difficulty with usual activities’ rather than just usual activities, if this is the outcome measure.
Response
We have replaced “usual activities” with “difficulty with usual activities” as suggested. Please see p9 line 32.
Some of the factors require clearer description. For example does ‘age’ refer to the age of the patient or to the presence of people of these ages in the household ?
Response:
Age refers to the age of the patient included in the research. We have edited the variable labels in the tables throughout the manuscript to clarify this.
P 9 line 3 and Table 4. The association with household size is unclear – as the number of household members or reference value is not mentioned. It is unclear what is meant by ‘time for drug procurement’ – which appears to be the term ‘duration of appointment for drug supplies’ in the table. This might be clearer by using wording such as ‘frequency of appointments for collecting drug supply’.
Response
Thank you for pointing these out. As for the number of household members, there is no reference value. Regarding the numeric variable, the interpretation follows: “for one unit increase in household size leads to a 5% increase in pain/discomfort in TB patients”. As for ‘time for drug procurement’, we agreed to replace it with ‘frequency of appointments for drug supply’. You can see changes throughout the manuscript.
P 9 line 8 provides a somewhat clearer statement of what is meant by ‘household size’ in relation to association with anxiety, although the clinical significance is unclear.
Response
Thank you for your insightful remark. We made changes to make a more explicit statement” For one unit increase in the household size, the odds of reporting pain and discomfort was 1.05 times greater (OR=1.05; CI: 1.00-1.11, p<0.05), and patients who were clinically diagnosed were 93% more likely to have pain and discomfort than those diagnosed bacteriologically (OR=1.93, CI: 1.06-3.54, p=0.031)…”. Please see p10 from lines 1 to 7.
P 10 Table 6 does not provide any figures for the factor ‘distance between site and residence’ although the text (line 5) refers to an AOR of 1.03, which, in the table 6, is provided for a duration of appointment of 15-30 days. The figure for duration of treatment of > 30 days is not provided in Table 6 although quoted in the text at line 7; nor is the data for extrapulmonary TB.
Response
Thank you for your excellent observations. We agreed, and we have made revisions to make the table coherent with the statement, “Regarding stress, as measured by IES-R, for one unit increase in the household size, the odds of reporting stress was 1.10 times greater (OR=1.08, CI: 1.03, 1.14, p=0.002). Non-co-infected TB patients were 55% less likely to be stressed than co-infected ones (OR=0.45, CI: 0.29-0.71, p<0.001). For an increase of one unit in the distance from patients’ residence to DTCs, the odds of anxiety were 1.03 times greater (OR=1.03, CI: 1.01-1.05, p<0.001). Similarly, patients whose frequency of appointments for drug supply was between 15- 30 days had 65% less likely to be stressed than those whose frequency of appointments for drug supply was ≤14 days (OR=0.35, CI:0.14,0.75, p=0.011). Likewise, patients whose frequency of appointments for drug supply was >30 days had 99,94% less likely to experience stress than those whose frequency of appointments for drug supply was ≤14 days (OR=0.06, CI: 0.00, 0.30, p=0.007; (Table 7)”. Please see p11 from lines 9 to 16
Qualitative findings
This section provides a clear description of some interesting findings. However, more information on the restrictions and measures introduced to control covid 19 in the settings section would assist the reader in understanding how these might impact on patients with TB.
Response
We agree with this comment and have added additional information to the introduction section of the paper. Please refer to our response to a similar suggestion from Reviewer 1 above.
Discussion
The reports of pain, discomfort, anxiety and depression by TB patients in the questionnaire are not compared to reported levels prior to the covid 19 pandemic, or to persons who do not have TB, and thus it is not possible to determine how these might have changed as a result of covid 19, or how these might compare with healthy individuals. The wording in the discussion needs to reflect some caution in interpretation.
Response
We agree that due to the lack of comparator, we are unable to definitively state what impact the COVID-19 pandemic has had on the levels of HRQoL. Nonetheless, we believe that the information from the qualitative interviews does enable some assessment of the impact of covid 19, although some of the comments (for example those related to livelihood) may well also apply to persons without TB. We have added some text in the discussion section to acknowledge this fact (Lines 19 – 25, page 15).
Caution is needed in the interpretation of the quality of life scale results, as the survey does not enable an analysis of the causative factors that might contribute to the differences among scales for different factors. For example, it is unclear whether the higher level of problems with daily activities or pain and discomfort for the elderly refer to problems of the elderly or problems of the household in managing elderly members. The discussion seems to confuse issues of the clinical management of elderly TB patients, together with issues of managing elderly household members.
The comments on p 13 regarding caution in interpretation of the findings are important and helpful.
Response
Thank for pointing out this. We have incorporated more clarifications as follow: “However, we must be aware that the management of diseases in the elderly both in the context or outside of COVID-19 presents a complexity in the Guinean context. Indeed, we lack suitable services to take care of vulnerable people. This is why, whenever we are faced with stressful situations, these elderly people are the first to feel the brunt». In relation to our inability to identify causative factors, please see our response to the comment above. Please see P14 from lines 24 to 28.
Submission Date
21 July 2022
Date of this review
08 Aug 2022 03:34:41
Round 2
Reviewer 3 Report
The authors have satisfactorily addressed all my comments on the original version, and have significantly strengthened the paper. Where comparison information is not available, this has been indicated.